# COVID-19 Vaccine Intention among Rural Residents in Mexico: Validation of a Questionnaire

**DOI:** 10.3390/vaccines9090952

**Published:** 2021-08-26

**Authors:** Hector S. Alvarez-Manzo, Rafael Badillo-Davila, Alejandro Olaya-Gomez, Barbara Gonzalez-de-Cossio-Tello, Rafael Cardoso-Arias, Emilio S. Gamboa-Balzaretti, Carlos D. Baranzini-Rogel, Gabriel Garcia-Garnica, Luis E. Hernandez-Corrales, Carlos A. Linares-Koloffon

**Affiliations:** 1Department of Molecular Microbiology and Immunology, Johns Hopkins Bloomberg School of Public Health, Baltimore, MD 21205, USA; 2Investigación Comunitaria e Implementación en Salud A.C., Mexico City 01900, Mexico; rafa4554@gmail.com (R.B.-D.); barbara.bgdc@gmail.com (B.G.-d.-C.-T.); rcardoso@mas.org.mx (R.C.-A.); clinareskoloffon@salud.unm.edu (C.A.L.-K.); 3Clínica MAS, Medicina y Asistencia Social A.C., Tlapa 41304, Mexico; esgb.1995@gmail.com (E.S.G.-B.); carlos_baranzini4@hotmail.com (C.D.B.-R.); gagg1710@hotmail.com (G.G.-G.); eliuduisa@hotmail.com (L.E.H.-C.); 4Department of Statistics, University of Washington, Seattle, WA 98195, USA; olaya19@uw.edu; 5School of Medicine, Universidad Panamericana, Mexico City 03920, Mexico; 6College of Population Health, University of New Mexico, Albuquerque, NM 87131, USA

**Keywords:** COVID-19, COVID-19 vaccine, vaccine intention, vaccine hesitancy, questionnaire validation, rural population, pilot study, Mexico, Latin America

## Abstract

The COVID-19 pandemic forced the scientific community and the pharmaceutical industry to develop new vaccines, in an attempt to reach herd immunity and stop the SARS-CoV-2 from spreading. However, to ensure vaccination among the general population, COVID-19 vaccine intention must be measured. So far, no studies have focused on rural residents in Latin America, which represent approximately 20% of the population of this geographical region. In this study, we present the validation of a self-developed questionnaire, which was validated in a pilot study with 40 Spanish-speaking Mexican rural residents in the state of Guerrero, Mexico. In this study, we describe the chronological validation of the questionnaire, including the assessment of its internal consistency and temporal reliability, which we measured with the Cronbach’s alpha and Spearman’s rank correlation coefficient, respectively. After the psychometrical analysis, we were able to validate a 20-item questionnaire, which intends to assess vaccine intention among the rural population. Aiming to develop a comprehensive policy and vaccination strategies, we hope this instrument provides valuable insight regarding COVID-19 vaccination willingness across rural communities in Mexico and Latin America. Finally, if we want to reach worldwide herd immunity, it is important to understand rural residents’ position towards COVID-19 vaccination.

## 1. Introduction

In December 2019, the World Health Organization (WHO) reported an outbreak of pneumonia associated with a severe acute respiratory distress syndrome (SARS) of unknown etiology in the City of Wuhan, China [1]. Later, the etiological agent was identified as a virus of the *Coronaviridae* family: SARS-CoV-2 [1]. Due to the worldwide spreading of this virus, on 11 March 2020 the WHO declared it as a pandemic [1]. As of 7 August 2021, the Johns Hopkins University and the WHO have reported that SARS-CoV-2 has infected more than 200 million individuals worldwide with more than 4.2 million deaths attributed to COVID-19 [2,3]. In order to contain the pandemic, global research has mobilized rapidly, resulting in new advances in basic and clinical research, mainly in the development of vaccines that may prevent infection by SARS-CoV-2 and the mortality associated with COVID-19 disease [4,5]. In December 2020, after demonstrating 95% effectiveness against COVID-19 infection [6], the United States Food and Drug Administration approved the emergency use of the COVID-19 vaccine produced by Pfizer-BioNTech [7]. Since then, multiple vaccine candidates have undergone emergency use clearance among different countries, leading to more than 4.3 billion vaccine doses administered worldwide up to 7 August 2021 [3].

Worldwide COVID-19 vaccine intention is variable between countries. Malik et al. [8] reported a 67% COVID-19 vaccine intention in a representative sample of the US population, however this result shows variability when demographic characteristics such as age, race, level of education and employment status are taken into account. Lazarus et al. [9] assessed the level of intention among 19 countries, showing that the level of it varies widely depending of the country (i.e., COVID-19 vaccine intention ranges from 54.85% in Russia, to 88.62% in China). Recent data also shows that people living in rural areas might be less prone to be vaccinated against COVID-19, and it is estimated that approximately 30% of the US rural population will either completely reject COVID-19 vaccination or will get vaccinated only if required to attend work, school, or other activities [10].

Recent studies conducted in Latin America (LATAM) have already assessed COVID-19 vaccine intention in the general population [9,11,12]. Nevertheless, to the best of our knowledge there are no studies that specifically reviewed the level of COVID-19 vaccine intention among the rural population. The International Labour Organization estimates that in LATAM 123 million persons live in rural areas [13], where Mexico accounts for 27 million [14]. Considering that vaccination will determine the number of cases and deaths from COVID-19 disease in the future, it is crucial to explore the prevalence of intention to be vaccinated against COVID-19 in rural communities, which are characterized by low educative and socioeconomic levels [14,15], making them vulnerable to the COVID-19 pandemic [16]. In this study, we present the results of the validation of a self-developed questionnaire that assesses COVID-19 vaccine intention and perception among rural residents in the Mexican state of Guerrero.

## 2. Materials and Methods

### 2.1. Design of Questionnaire

Aiming to design a culturally appropriated questionnaire, we performed a literature review on preexisting studies and expert’s ideas and opinions on the assessment of COVID-19 vaccine acceptance and perception among the general population. Given the target population of this study, the questionnaire was developed in Spanish. In addition, a Spanish-speaking linguist reviewed the questionnaire to ensure the accurate phrasing of each item according to the social and educative level of the study population.

We initially drafted 45 items, and these were categorized in six sections: (1) Perception risk towards COVID-19 disease, (2) Perception of COVID-19 sources of information, (3) Positive perception towards vaccination, (4) Positive perception towards COVID-19 vaccination, (5) Negative perception towards COVID-19 vaccination, and (6) Personal beliefs as barriers to vaccination.

#### 2.1.1. Perception Risk towards COVID-19 Disease

In this section, we sought to assess the awareness of participants towards COVID-19 disease, as it is a factor known to be associated with positive preventive behavior, (i.e., vaccination intention) [17,18]. To determine this, we questioned participants’ general perception towards the COVID-19 disease through a series of items that interrogate their attitudes during specific scenarios.

#### 2.1.2. Perception of COVID-10 Sources of Information

The most common sources of COVID-19-related information that have been utilized by the general population are the internet media, traditional media (i.e., television), family members, co-workers, friends, and health professionals [19]. Thus, in this section we sought to identify the perception that participants have regarding the reliability of information sources, so we could determine the origin of misinformation that is possibly responsible for COVID-19 vaccine hesitancy.

#### 2.1.3. Positive Perception towards Vaccination

Considering the key barrier categories from the “Model of Determinants of Vaccine Hesitancy” by the SAGE working group [20,21], we intended to address the patients’ viewpoint concerning vaccination through a series of items that question their knowledge regarding vaccination and the potential barriers responsible for vaccination reluctance.

#### 2.1.4. Positive Perception towards COVID-19 Vaccination

Based on previous studies that assessed the perception of COVID-19 vaccination [8,9,11], we designed items that evaluated factors influencing COVID-19 vaccine positive opinion.

#### 2.1.5. Negative Perception towards COVID-19 Vaccination

The causes behind vaccine hesitancy are somehow various and ambiguous, ranging from sociodemographic up to psychological factors, such as mistrust towards healthcare professionals and political authorities [22]. This section questions probable barriers responsible for the negative impact on the intention to vaccinate against COVID-19. Participants were asked about the controversy concerning the vaccine efficacy, safety and possible side effects.

#### 2.1.6. Personal Beliefs as Barriers to Vaccination

Based on the negative association between religiosity and COVID-19 vaccination showed by a previous study [23], and previous reports that religious leaders could negatively impact the vaccination process by distorting the available information about the pandemic [24], we sought to evaluate the influence on COVID-19 vaccination intention of participants’ personal beliefs.

### 2.2. Content Validity

A panel of seven professionals with expertise in epidemiologic and community research assessed the qualitative and quantitative validity of the questionnaire. Furthermore, we gathered comments on wording and vocabulary from the expert’s panel and modified items accordingly. Then, the quantitative validity was measured by asking the experts to grade the relevance of each item with a 4-point Likert scale (“not relevant”, “something relevant”, “relevant”, or “very relevant”). Answers for each item were coded as “0” if the answer of the expert was “not relevant” or “something relevant”, and as “1” if the answer of the expert was “relevant” or “very relevant”, and the content validity index for individual items (CVI-I) was calculated as previously described [25,26]. The threshold for the CVI-I for keeping an individual item or discarding it from the questionnaire was set at 0.8, meaning that at least 6 of the 7 experts had to grade each item as “relevant” or “very relevant”. After receiving the experts’ comments and grading, 4 items had to be eliminated because of a CVI-I < 0.8. Moreover, items were further modified based on the expert’s comments regarding phrasing and vocabulary used, to ensure understanding of the items by the population of study.

### 2.3. Internal Consistency and Reliability

Once items were discarded based on the assessment of the experts’ panel, a pilot study was conducted in-person with 40 participants. During the study period, COVID-19 vaccination was available exclusively for healthcare professionals and people ≥ 60 years old. The inclusion criteria were (a) ≥18 years old, (b) Spanish as a primary or native language, and (c) participants must be Mexican citizens and live in the mountain region of the Mexican state of Guerrero (see Appendix B for the geographic detail). The exclusion criteria were (a) the participant is (or was in case of retirement) a healthcare worker (physician, nurse, emergency medical technicians) and (b) the participant has already been (partially or fully) vaccinated against COVID-19. Google Forms was utilized to apply the 41-item questionnaire in a healthcare setting by four primary care physicians at “Clínica MAS”, a primary care clinic located in the city of Tlapa in the Mexican state of Guerrero. In this study, participants had to choose for each item with a 5-point Likert scale if they agreed or disagreed with each statement (“strongly disagree”, “disagree”, “neither agree nor disagree”, “agree”, or “strongly agree”).

For evaluating the internal consistency for each category of the questionnaire, we calculated Cronbach’s alpha for each section, and the threshold was set at 0.7 as previously described [26,27]. To measure the questionnaire’s reliability, the test–retest method was utilized. In this study, questionnaires were applied twice to each participant with a difference of 1–2 weeks between the first and the second interview. The Spearman’s rank correlation coefficient between the first and the second questionnaire responses (for each subject) was calculated for each item and the threshold to discard individual items was set at 0.6, which reflects a strong correlation and ensures the repeatability of the questionnaire [28]. We decided to use the Spearman’s rho, since it is used for measuring the monotonic correlation between two variables; this is less restrictive than the linear correlation measured by Pearson’s correlation coefficient. In this sense, it is better suited for measuring the correlation between ordinal variables (such as a Likert scale) as it helps to understand if high ranks of one variable are associated with high ranks of another variable (correlation close to 1), with low ranks of the other variable (correlation close to −1) or if there is no association between variables (correlation close to 0).

### 2.4. Statistical Analysis and Ethical Standards

Cronbach’s alpha and Spearman’s rank correlation coefficient were calculated with “R” software. The final questionnaire in Spanish can be found in the Appendix A (Appendix A), as well as the back-translation into English (Appendix A). In addition, the raw data of the 40 participants (Appendix A) can also be found in the Appendix A.

This study was approved by the Ethics Committee of the “Fundación Clínica Médica Sur A.C.”, based in Mexico City, with the approval number 2021-EXT-554 and performed in accordance with the Declaration of Helsinki. Written informed consent was required from all participants and data obtained were handled confidentially.

## 3. Results

We interviewed a total of 95 participants from 11 March to 13 May, from which 40 participants came back to the clinic to complete the second questionnaire 1–2 weeks after the first interview. Participants’ age ranged from 18 to 57 years old, and 70% of them were women. The results from the first interview were used for evaluating the internal consistency of the questionnaire, while the information gathered from the second interview was utilized for measuring the temporal reliability via the test–retest method.

The statistical analysis consisted of the computation of Cronbach’s alpha for internal consistency within the first six sections of the questionnaires, and of Spearman’s rank correlation for temporal reliability for each section and item. Moreover, the answers to items 10–12 and 20–22 were coded in the opposite direction as they were answered. This was conducted with the objective of keeping the same sense of answers in Sections 2 and 3, respectively. Table 1 shows the results for internal consistency and temporal reliability for each section of the test. Each subsection was designed to measure different elements related to the COVID-19 vaccine, and what we were looking for is a high alpha for each section. It is important to notice that both the first (“Perception risk towards COVID-19 disease”) and second (“Perception of COVID-19 sources of information”) sections of the questionnaire had a Cronbach’s alpha < 0.7, hence these sections were completely excluded (items 1–14). In comparison, Sections 3–6 had a Cronbach’s alpha > 0.7, thus these were kept in the final questionnaire.

All sections were found to have a Spearman’s correlation coefficient greater than 0.6; in fact, the questionnaire as a whole had a 0.86 Spearman’s rank correlation. This means that the test had a high temporal reliability. In addition, a Spearman’s rank correlation coefficient was also calculated for each item individually, and items 15, 16, 17, 24, 26, 31 and 36 also had to be excluded from the final questionnaire, since their individual Spearman’s rank correlation coefficient was <0.6; for the rest of the items this value ≥ 0.6. The results for the final questionnaire are shown in Table 2; as it can be seen, all the remaining sections have a Cronbach’s alpha and a Spearman’s rank correlation higher than their respective thresholds, which ensures internal validity and temporal reliability for the final questionnaire (see Appendix A for individual values for each item).

In total, 21 items had to be eliminated from the final questionnaire either because of low internal consistency among items in a section (Sections 1 and 2), or because of a low temporal reliability (items 15, 16, 17, 24, 26, 31 and 36), leaving us a 20-item questionnaire. Table 3 presents examples of the items (translated into English) of the different sections of the final questionnaire.

## 4. Discussion

Seventeen months after the start of the COVID-19 pandemic, Brazil, Argentina, and Colombia are located in the top 10 countries with the most COVID-19 cases. More dramatically, Brazil, Mexico, and Peru are in the top five of total deaths directly caused by the COVID-19 disease, despite none of them being one of the five most populated countries worldwide [3]. These facts indicate that the LATAM region has been deeply affected by the COVID-19 pandemic and that vaccine intention in this geographical area needs to be assessed. Moreover, it has been reported that the median R_0_ (basic reproduction number) value of SARS-CoV-2 is 5.7 (95% CI 3.8–8.9), which means that approximately 82% of the world population will need to be immune either by COVID-19 vaccination or SARS-CoV-2 infection, to achieve herd immunity and stop the chain of transmission [29]. If we take into consideration that approximately 20% of LATAM inhabitants are considered rural residents [13], it is necessary to focus on this population if public health and government officials want to reach herd immunity. Thus, we developed and validated a questionnaire (see Appendix A for full questionnaire in Spanish and English translation, respectively) that aims to identify COVID-19 vaccine intention and the factors related among Spanish-speaking Mexican rural residents.

As of 7 August 2021, 39% of the Mexican population has been at least partially vaccinated [30], hence there is still a long way until reaching herd immunity in Mexico. Furthermore, COVID-19 vaccination in LATAM varies across nations dramatically, and vaccination rates range from <20% in the case of Guatemala, Honduras, Nicaragua and Venezuela, to >70% in Chile, Puerto Rico and Uruguay (Figure 1). We recognize that COVID-19 vaccines have been available since the beginning of 2021 around the globe. Nevertheless, equitable distribution of COVID-19 vaccines is a worldwide problem [31], and as observed in Figure 1, LATAM is not the exception. Since Mexico shares a historical and cultural background with the rest of LATAM, we hope that this questionnaire provides a validated instrument to other Spanish-speaking countries in this geographical region to study their rural population, intended to improve their policy and vaccination strategies, especially those with vaccination rates below 20% (Figure 1) such as Nicaragua and Venezuela.

The reason for the low vaccination rates in some LATAM countries needs to be addressed in future studies, although it is highly probable that the main barriers to vaccination are the lack of economic resources to secure COVID-19 vaccines, and the inability to implement vaccination because of supply chain-related problems [32,33]. However, attitudinal barriers have also been described as interfering with the person’s willingness to seek a vaccination service provided by the government, in this case for getting vaccinated against COVID-19 [33]. Previous studies have already investigated the role that the perceived risk [34,35,36,37], mistrust [34,35,36,38,39], misinformation [36,40,41], and personal beliefs [23,36] towards COVID-19 disease and COVID-19 vaccination play among the general population, and have established that these factors are important to be taken into account for ensuring a successful COVID-19 vaccination program [33]. Hence, the items in this questionnaire are focused on identifying the attitudinal barriers previously described that could be interfering with the intention to vaccinate rural residents, so that a better vaccination strategy that targets vaccine-hesitant people can be designed and implemented.

We are enthusiastic about the validation of this questionnaire and hope that this tool can be applied in a future research project that takes a representative Mexican rural population, to address the question of whether Mexican rural residents are willing or hesitant to be vaccinated against COVID-19. However, despite careful questionnaire design, this study also comes with some limitations. Our questionnaire was designed in Spanish for Spanish-speaking people. Nevertheless, some of the rural residents in Mexico do not speak Spanish and instead speak any of the more than 60 indigenous languages that are spoken across the country. It is estimated by the National Institute of Statistics and Geography that approximately 6.7 million Mexicans speak an indigenous language [42], and most of them live in rural areas. Thus, since approximately 25% of rural residents in Mexico do not speak Spanish as their native language, there is an important subset of the Mexican rural population that cannot be studied with our questionnaire. Instead, translation of this questionnaire into indigenous languages and subsequent validation could be carried out in future studies, to assess COVID-19 vaccine intention among the non-Spanish-speaking rural population.

## 5. Conclusions

In this study, we present the validation of a self-developed 20-item questionnaire that seeks to address COVID-19 vaccine intention in a Mexican rural population. To the best of our knowledge, this is the first study that specifically develops a questionnaire to assess vaccine intention among rural residents in LATAM. Thus, we hope that the tool we are providing helps future studies in Mexico, but also in other Spanish-speaking countries in LATAM, to understand rural residents’ intention to get vaccinated against COVID-19, and the factors that could be determining their willingness or reluctance to be immunized.

## Figures and Tables

**Figure 1 vaccines-09-00952-f001:**
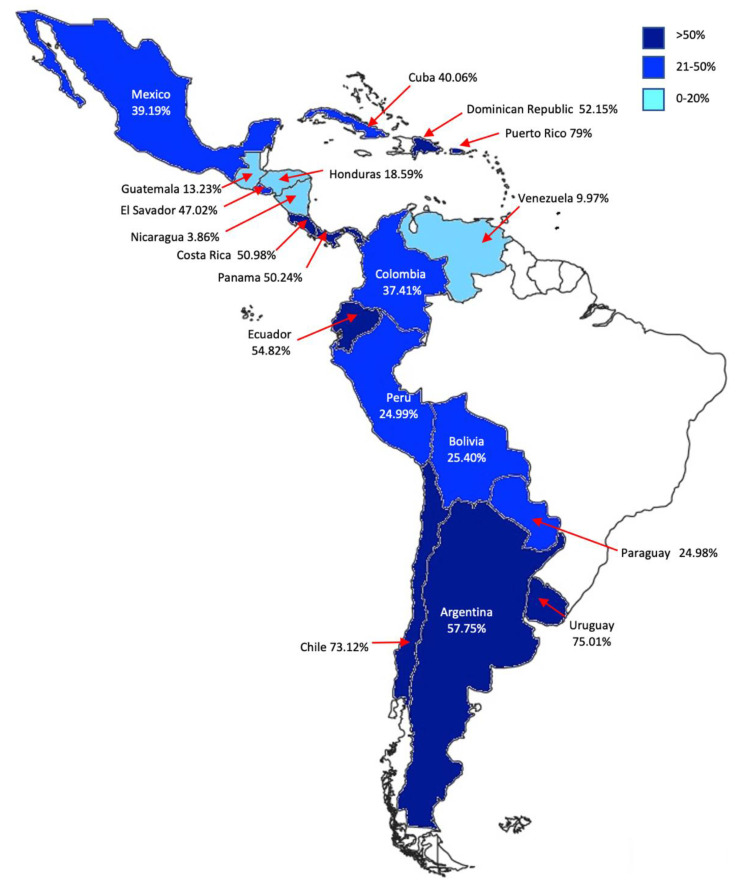
Percentage of the population vaccinated with at least one dose of a COVID-19 vaccine in Spanish-speaking countries in Latin America as of 7 August 2021, based on the vaccination data provided by Mathieu et al. [30].

**Table 1 vaccines-09-00952-t001:** Cronbach’s alpha and Spearman’s rho by section of the preliminary questionnaire.

Section ^1^	Cronbach’s Alpha	Spearman’s Rho
1 ^2^	0.53	0.64
2 ^2^	−0.25	0.85
3	0.78	0.65
4	0.82	0.61
5	0.88	0.67
6	0.79	0.81
Entire questionnaire	N/A	0.86

^1^ Names of the sections can be found in “2.1 Design of Questionnaire”. ^2^ Sections 1 and 2 were excluded from the final questionnaire due to a low Cronbach’s alpha value. N/A means “not applicable”.

**Table 2 vaccines-09-00952-t002:** Cronbach’s alpha and Spearman’s rho by section of the final questionnaire.

Section ^1^	Cronbach’s Alpha	Spearman’s Rho	Items Removed
3	0.76	0.71	15, 16 and 17
4	0.72	0.66	24 and 26
5	0.90	0.68	31 and 36
6	0.79	0.81	None
Entire questionnaire	N/A	0.90	15, 16, 17, 24, 26, 31 and 36

^1^ Names of the sections can be found in “2.1 Design of Questionnaire”. N/A means “not applicable”.

**Table 3 vaccines-09-00952-t003:** Examples of the items by section of the final questionnaire.

Section ^1^	Item ^2^
3	I trust in the advice of the medical and nursing staff when they tell me I need to get vaccinated.
4	As soon as I have the opportunity, I will get vaccinated against COVID-19.
5	I will not get vaccinated against COVID-19 because the vaccine is not safe.
6	I am going to get sick no matter what I do.

^1^ Names of the sections can be found in “2.1 Design of Questionnaire”. ^2^ For all items, participants had to decide with a 5-point Likert scale their level of agreement with the statement.

## Data Availability

The data presented in this study are available in the Appendix A.

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
