# Peer review of "COVID-19 Vaccine Intention among Rural Residents in Mexico: Validation of a Questionnaire"

_vaccines, 2021, doi:10.3390/vaccines9090952_

Round 1
Reviewer 1 Report
The authors reported the intention of Covid-19 vaccination in 95 adult rural residents in Mexico, based on the survey with a “self-designed” questionnaire. Although the authors considered that their 20-item questionnaire may provide valuable insight about the vaccination intention in rural areas in Latin America, the authors do not provide convincing evidence to support the conclusion. Numerous surveys on the Covid-19 vaccination intention in various populations from different countries, including in Mexico, have been reported. The questions in the designed questionnaire in this manuscript are generally similar to those reported in published surveys. The self-designed questionnaire does not add novel critical questions. In addition, there are several other critical issues in this study.
- A sample size of 95 adult rural residents participated in this study was too small to get a valid conclusion.
- The study participants enrolled from the Mountain region of the Mexican state of Guerrero can just represent the population in this region, but cannot represent the population in Mexico, and can say nothing of the populations in Latin America.
- As the authors mentioned in the manuscript, that 3.5 billion vaccine doses were administered worldwide up to July 2021 (line 49), survey about Covid-19 vaccination intention now is out of date.
Other comments:
- This study just investigated the intention of Covid-19 vaccination. As Covid-19 vaccines are available now, the “Covid-19 vaccine acceptance” used in the manuscript may be mistakenly considered the actual acceptance of Covid-19 vaccination. It is not adequate to use “Covid-19 vaccine acceptance”.
- A reference(s) should be cited to validate the 1st sentence of the Introduction.
- Line 38: “its’” is not correct.
- Table 1: what is the logic to present the subjects’ weight and height? Did the authors consider whether or not getting vaccinated against Covid-19 is related with weight and height?
5. The Appendix A is not presented.
Reviewer 2 Report
Thanks Authors to choose MPDI, Vaccines to publish their manuscript.
Introduction well describe the topic.
Material and Methods together with supplementary material make you perfectly understand the conducted experiment.
Results and statistical l-analysis further elucidate the manuscript.
Institutional Review Board Statement is ok.
Nothing to do.
Reviewer 3 Report
This article studies the COVID-19 vaccine acceptance among rural residents in Mexico. So far, no elated studies have focused on rural residents in Latin America, representing approximately 20% of the population of this geographical region. This article has made a good start. This article claims the design of the questionnaire that initially drafted 45 items. Moreover, these items were categorized into six sections. However, the current form is too simple and short. The authors should provide more detailed information, such as the design of the questionnaire.
Round 2
Reviewer 1 Report
Hundreds of articles on the intention of Covid-19 vaccination have been published throughout the world. Before the availability of Covid-19 vaccines, such study is meaningful. However, as Covid-19 vaccines became available more than half year, I really do not understand what the meaning of a study on the survey of Covid-19 vaccination intention is.
Another question is what the difference between the authors’ self-developed questionnaire and reported ones. Actually, most of the reported questionnaires have similar questions.
Thus, although the authors answered the comments, I am not persuaded to suggest the acceptance of this manuscript.
Author Response
It is true that COVID-19 vaccines are now in theory available worldwide. However, there is a huge difference between the theory, and what is actually happening. Probably the best example of what we are trying to illustrate is Nicaragua. Although COVID-19 vaccines are available worldwide since the beginning of 2021, Nicaragua has only vaccinated (either partially or fully) 3.9% of the population as of August 7th, which corresponds to an approximate of 250,000 people. In comparison, Denmark which has a similar population, has already vaccinated 74% of their population. Thus, the fact that it is in theory "available" worldwide, does not mean that it really is available in all countries evenly. Equitable distribution of COVID-19 vaccines, as well as health infrastructure to allocate and distribute vaccines, is an important problem worldwide [1] and Latin America is not the exception as can be seen in Figure 1. This point is of course out of the scope of our study, however it is the justification why the validation of our questionnaire is meaningful and relevant, and why our study although "six months late", is still meaningful. This key-point has been emphasized now in lines 321-324.
Moreover, the main differences between our study and previous studies are the following:
1. Validation of the questionnaire (most questionnaires and surveys in previous studies are not previously validated).
2. Focused on rural population (no previous study has focused on this group).
3. Questionnaire was designed in Spanish (no previous study has designed a questionnaire in Spanish).
The importance of these key-points, as well as the justification why these are important, are detailed in the previous cover letter sent after the Review Report in Round 1, and were already included in the previous manuscript-version submitted.
Although we were not able to persuade you from the importance of our study, we truly appreciate your insight. We do believe that the corrections we did based on your comments have improved the quality of our article.
References:
1. Burki, T. Equitable distribution of COVID-19 vaccines. Lancet Infect Dis 2021; 21(1):33-4.
Reviewer 3 Report
The revised article has addressed and solved all of my concerns. Hence, I think that the revised article can be accepted for publication in this journal.
Author Response
We appreciate the comments in round 1 that have certainly improved the quality of the paper.